# A MENTION-PAIR MODEL OF ANNOTATION WITH NONPARAMETRIC USER COMMUNITIES

## ABSTRACT

The availability of large datasets is essential for progress in coreference and other areas of NLP. Crowdsourcing has proven a viable alternative to expert annotation, offering similar quality for better scalability. However, crowdsourcing require adjudication, and most models of annotation focus on classification tasks where the set of classes is predetermined. This restriction does not apply to anaphoric annotation, where coders relate markables to coreference chains whose number cannot be predefined. This gap was recently covered with the introduction of a mention pair model of anaphoric annotation (MPA). In this work we extend MPA to alleviate the effects of sparsity inherent in some crowdsourcing environments. Specifically, we use a nonparametric partially pooled structure (based on a stick breaking process), fitting jointly with the ability of the annotators hierarchical community profiles. The individual estimates can thus be improved using information about the community when the data is scarce. We show, using a recently published large-scale crowdsourced anaphora dataset, that the proposed model performs better than its unpooled counterpart in conditions of sparsity, and on par when enough observations are available. The model is thus more resilient to different crowdsourcing setups, and, further provides insights into the community of workers. The model is also flexible enough to be used in standard annotation tasks for classification where it registers on par performance with the state of the art.

## 1 INTRODUCTION

Identifying and resolving anaphoric reference to discourse entities, a task known in NLP as *coreference resolution*, has long been considered a core aspect of language interpretation (Poesio et al., 2016). Ever since the MUC evaluation campaign in the 1990s (Grishman & Sundheim, 1995; Chinchor, 1998), larger and richer datasets have been made available, pushing the state of the art to new heights. In the last few years, the ONTONOTES corpus (Pradhan et al., 2007; Weischedel et al., 2011), used for the CONLL 2011 and 2012 shared tasks (Pradhan et al., 2012), has become the de facto standard resource for coreference resolution research (Fernandes et al., 2014; Björkelund & Kuhn, 2014; Martschat & Strube, 2015; Clark & Manning, 2015; 2016a;b; Lee et al., 2017; 2018). The corpus was hand-annotated by experts, and remained the largest available dataset up until the recent publication of PRECO (Chen et al., 2018). But there are still many languages and domains for which no such resources are available, or where the annotation scheme is limited (e.g., the lack of singletons in ONTONOTES, no expletives in either PRECO nor ONTONOTES).

Annotating data on the scale required to train state of the art systems using traditional expert annotation can quickly get unaffordable. But in recent years crowdsourcing has proved a viable alternative to expert annotation, with studies indicating that expert-level quality can be achieved with much lower costs (Snow et al., 2008; Raykar et al., 2010). Crowdsourced data however require aggregation methods to choose the most likely label(s) among the interpretations provided by the crowd. Past research suggests probabilistic models of annotation are one of the most promising approaches to aggregation (Dawid & Skene, 1979; Carpenter, 2008; Whitehill et al., 2009; Raykar et al., 2010; Hovy et al., 2013; Quoc Viet Hung et al., 2013; Sheshadri & Lease, 2013; Passonneau & Carpenter, 2014; Venanzi et al., 2014; Kamar et al., 2015; Paun et al., 2018a). These models offer a rich framework of interpretation and can employ distinct prior and likelihood structures (pooled, unpooled, and partially pooled) and a diverse set of effects (annotator ability, item difficulty).

**Motivation** Most work on models of annotation assume the set of classes the annotators can choose from is fixed across the annotated items, an aspect not appropriate for anaphoric annotation where coders relate markables to anaphoric chains. Recently, Paun et al. (2018b) developed a probabilis-

tic model able to aggregate crowdsourced anaphoric annotations. The model was later applied to adjudicate the interpretations from the *Phrase Detectives 2* corpus with comparable quality to that of expert annotators (Poesio et al., 2019). The model of Paun et al. (2018b) assumes an unpooled structure, i.e., it models individual annotator parameters. Such models typically require a larger number of observations to properly estimate the ability profile of the coders. In a crowdsourcing environment this requirement may not always be satisfied, e.g., in the intial stages of a crowdsourcing campaign, or in the commonly encountered scenario where the workload of the annotators resembles a power law curve (Ipeirotis, 2010; Chamberlain, 2016); both examples describe a sparse data environment which may prove difficult to handle for an unpooled model. One intuitive solution to this problem is to exploit the similarities found in the behaviour of the annotators. Simpson et al. (2011; 2013) identified, after fitting a model of annotation, distinctive clusters in the ability of the workers; more generally, typical annotator communities found in a crowdsourcing setup include spammers, adversarial, biased, average or high quality players. Knowledge of these communities would allow to regularize the ability of the annotators towards the profile of the community they are part of. This partially pooled structure can prove effective in conditions of sparsity where there are not enough observations to accurately estimate the ability of the annotators in isolation. The level of pooling would be dictated by the data, such that, when enough observations are gathered, the partially pooled and the unpooled models would perform similarly.

**Contributions** In this work we extend the unpooled mention pair model of annotation (MPA) proposed by Paun et al. (2018b) with hierarchical communities of annotators. We let the number of communities grow with the data, a flexibility that we achieve using a Dirichlet process mixture based on a stick-breaking representation of the underlying Dirichlet process. We conduct the evaluation on the *Phrase Detectives 2* corpus, in various levels of sparsity, assessing the accuracy of the inferred mention pairs, the quality of the post-hoc constructed silver chains, and the viability of using silver chains as an alternative to expert-annotated chains when training a state of the art coreference system. We discuss the inferred community profiles of a few known spammers and honest players of the game used to collect the *Phrase Detectives 2* corpus. We conclude the evaluation with a performance check on traditional crowdsourcing datasets against several state of the art models.

## 2 A COMMUNITY MODEL FOR ANAPHORIC ANNOTATIONS

We use the same annotation scheme as Paun et al. (2018b): annotators mark mentions as discourse new if a new entity is being introduced into the discourse (represented internally as DN), with property for predicative noun phrases (PR), with non-referring for expletives (NR) and with discourse old if an already introduced entity is being mentioned, in which case the annotators must also specify the mention's most recent antecedent (DO(ante-id)). We also follow the same notation as in Paun et al. (2018b) and use the term label to refer to a given annotation (i.e., DN, DO(ante-id), PR, NR) and the term class to refer to a general category a given label belongs to (DN, DO, PR, NR).

### 2.1 MODEL SPECIFICATION

Similar to MPA, the model we propose – which we will henceforth refer to as COMMUNITYMPA – assumes a pre-processing step in which the mention-level annotations are transformed into a series of binary decisions with respect to each (distinct) candidate label. Both models assume a similar generative process of these decisions, although for COMMUNITYMPA we used a different parameterization to later accommodate the hierarchical structure:

- For every mention $i \in \{1, 2, ..., I\}$:
    - For every (distinct) candidate label $m \in \{1, 2, ..., M_i\}$:
        * Draw true label indicator $c_{i,m} \sim \text{Bern}(\pi_{z_{i,m}})$ [1]
        * For every position $n \in \{1, 2, ..., N_{i,m}\}$:
            · If $c_{i,m} = 1$ then draw decision based on the sensitivity of the annotator $y_{i,m,n} \sim \text{Bern}(\Upsilon(\alpha_{jj[i,m,n],z_{i,m}}))$ [2] [3]

---

[1] $z_{i,m}$ is the class of the m-th candidate label for mention i.

[2] $\Upsilon()$ is the standard logistic function.

[3] jj[i,m,n] returns the index of the annotator who made the n-th decision on the m-th label of mention i.

· Otherwise, draw decision based on annotator's specificity $y_{i,m,n} \sim \text{Bern}(1 - \Upsilon(\beta_{jj[i,m,n],z_{i,m}}))$

The annotators belong to different communities and have abilities that depend on the class of the mentions they annotate:

- For every annotator $j \in \{1, 2, ..., J\}$:
    - Draw a community $x_j \sim \text{Cat}(\Phi(\Delta))$ [4]
    - For every class $h \in \{1, 2, ..., K\}$:
        * Draw sensitivity $\alpha_{j,h} \sim \text{Normal}(\tau_{x_j,h}, \sigma_{x_j,h})$
        * Draw specificity $\beta_{j,h} \sim \text{Normal}(\nu_{x_j,h}, \Omega_{x_j,h})$

The communities serve as hierarchical priors on the ability of the annotators regularizing it towards their mean as strongly as evidenced by the data (effect captured by the variance parameters):

- For every community $r \in \{1, 2, ..., \infty\}$:
    - Draw a stick proportion $\Delta_r \sim \text{Beta}(1, b)$ [5]
    - For every class $h \in \{1, 2, ..., K\}$:
        * Draw mean sensitivity $\tau_{r,h} \sim \text{Normal}(d_0, d_1)$
        * Draw sensitivity variance $\sigma_{r,h} \sim \text{Inverse} - \text{Gamma}(e_0, e_1)$
        * Draw mean specificity $\nu_{r,h} \sim \text{Normal}(t_0, t_1)$
        * Draw specificity variance $\Omega_{r,h} \sim \text{Inverse} - \text{Gamma}(u_0, u_1)$

Finally, the model is completed with conjugate priors:

- For every class $h \in \{1, 2, ..., K\}$:
    - Draw class specific true label likelihood $\pi_h \sim \text{Beta}(a_0, a_1)$
- Draw a scale $b \sim \text{Gamma}(s_0, s_1)$ [6]

Both MPA and its extension COMMUNITYMPA function as components in a standard mention pair framework: each mention is assigned the most likely candidate label based on the posterior of the label indicators, and the coreference chains are built from the mention pair links.

## 2.2 PARAMETER ESTIMATION NOTES

We estimate the parameters of the proposed model using variational inference, which is deterministic, typically fast, and benefits from a clear convergence criterion (Blei et al., 2017).

We parameterized the model with conjugacy in mind, making sure the complete conditionals are part of the exponential family. In this case the corresponding variational distributions take the same form and have the natural parameters equal to the expected value (under the variational distribution) of the natural parameters of the complete conditionals (Blei & Jordan, 2006; Hoffman et al., 2013). Conjugacy was not directly obtained in those cases involving the standard logistic function; we addressed that using the bound from Jaakkola & Jordan (2000). Lastly, the stick breaking representation of the Dirichlet process nicely complies with conjugacy as well (Blei & Jordan, 2006). We approximate the infinite mixture of user communities with truncated variational distributions.

The derivations are somewhat standard in the machine learning literature and for space constrains are omitted from the main paper but included in Appendix A.

## 3 EVALUATION

We conducted the experiments on the recently made available *Phrase Detectives 2* corpus (Poesio et al., 2019). The dataset was annotated in a game with a purpose setting, where 98,67% of the

---

[4] $\Phi(\Delta)$ is a vector of stick lengths, where the length of the r-th stick is $\Phi_r(\Delta) = \Delta_r \prod_{r'=1}^{r-1}(1 - \Delta_{r'})$. The different stick lengths represent the prevalence of the communities.

[5] The length of the stick broken at that proportion gives the prevalence of the community.

[6] The scaling parameter affects the growth of the number of communities with the data.

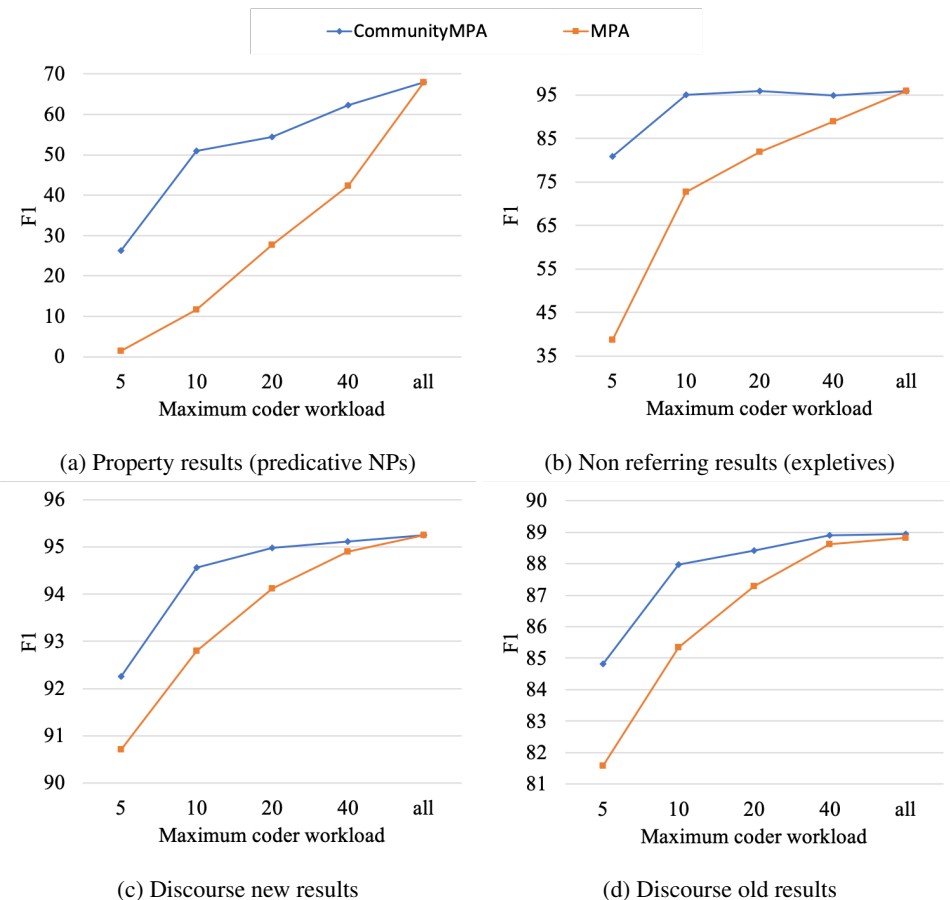

Figure 1: A per class evaluation of the inferred mention pairs matched against expert annotations. The "all" configuration uses the dataset as it is (does not alter the player workloads).

collected judgements come from players who produced more than 40 annotations each. To simulate a more sparse evaluation environment we break the larger player workloads into smaller batches which we further assume each was produced by a different player such that the workload of the players does not exceed a fixed threshold. Under this procedure the annotations are kept unchanged, offering a larger confidence when assessing whether the differences in performance between the partially-pooled COMMUNITYMPA model and its unpooled counterpart MPA come from the number of annotator observations that is required for each of these models to properly estimate their ability.

## 3.1 MENTION PAIR ACCURACY

This subsection presents the results for the agreement between the inferred mention pairs and the gold pairs. Both COMMUNITYMPA and MPA models were trained on the full *Phrase Detectives 2* corpus and evaluated on the expert-annotated subset. Both model implementations produce posterior point estimates for each candidate interpretation; we assign each mention the interpretation with the most mass under the posterior.

Figure 1 shows the results obtained by the models, for each class, under different workload configurations (i.e., maximum player workload). The trend lines clearly indicate a better performance of COMMUNITYMPA, across all classes. The gap in performance between the two models is the largest when the maximum number of observations (annotations) a player can have is capped at 5 and closes as the number of available observations increase, reaching on par performance when the dataset is used as it is (the "all" configuration). Remember that when the corpus is used as it is, almost all the data (98,67%) is produced by players with more than 40 annotations each, having plenty

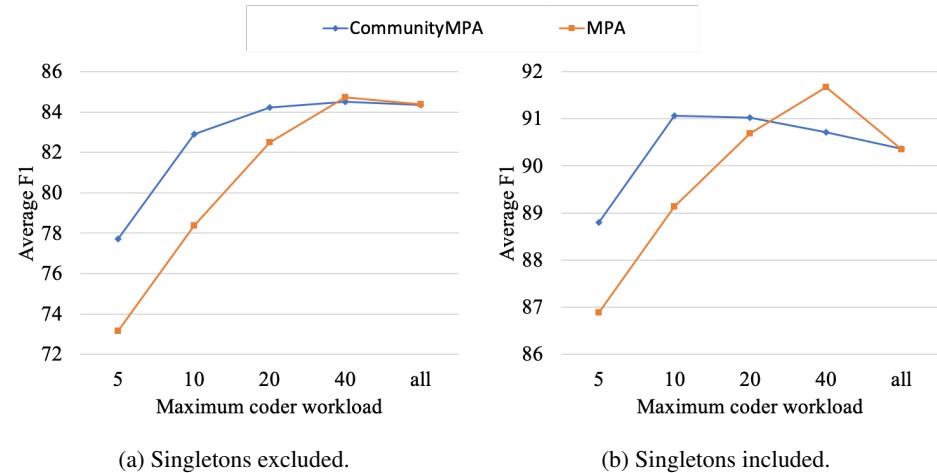

(a) Singletons excluded.          (b) Singletons included.

Figure 2: The quality of silver chains evaluated against gold chains.

of observations for an unpooled model like MPA to properly profile their ability. We saw however that MPA suffers in conditions of sparsity which the partially-pooled COMMUNITYMPA model alleviates through the hierarchical structure (the annotators' ability is pooled towards the ability of the community they are part of).

### 3.2 SILVER CHAIN QUALITY

We build the silver coreference chains from the inferred mention pairs by following the link structure inherent in the pairs. We assess the quality of the constructed chains by comparing them against gold (expert annotated) chains using standard coreference metrics. For this evaluation we used the scorer introduced by Poesio et al. (2018) to assess the quality of the chains both in a traditional CoNLL style (without singletons) and having singletons included into the evaluation.

Figure 2 shows the average F1 obtained when assessing the silver chains produced by the two models. The trend lines are similar to those from the mention pair evaluation: a better performance of COMMUNITYMPA that increases with the maximum player workload, registering the largest gap over MPA at the lowest number of observations considered in the evaluation, and on par performance when the dataset was used as it is. Again, we could see the benefits in low count data brought by the hierarchical structure of COMMUNITYMPA.

### 3.3 TRAINING ON SILVER CHAINS

In this Section we assess the viability of using silver chains as an alternative to expert-annotated chains when training a state of the art coreference system. We use the system used by Poesio et al. (2019), as other coreference systems cannot classify singletons and non referring expressions (i.e., expletives and predicative NPs), one of the unique characteristics of the *Phrase Detectives 2* corpus. We used as test data the gold chains that come with the corpus, while for training and development we used silver chains constructed a posteriori from aggregated mention pairs. We report the results obtained using the scorer introduced in the previous section which can handle both singletons and non-referring expressions in addition to the traditional CoNLL evaluation.

The results, presented in Figure 3, paint a similar picture to those from the previous evaluation sections: a better performance of the COMMUNITYMPA model in conditions of sparsity, with MPA closing the gap as more observations are allowed for each coder.

### 3.4 INFERRED USER COMMUNITIES

The *Phrase Detectives 2* dataset also includes an anonymized list of spammers and one with honest, well-established players from the game the corpus was collected with. We use these two lists to introduce and discuss some of the inferred communities these players were assigned to.

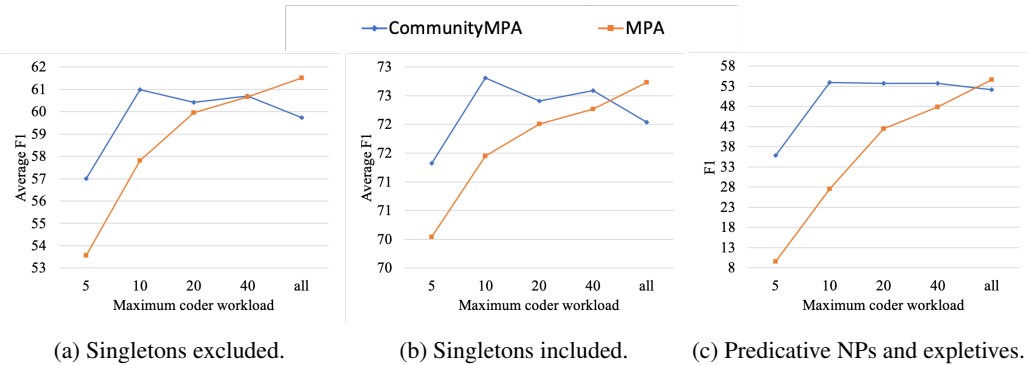

(a) Singletons excluded.    (b) Singletons included.    (c) Predicative NPs and expletives.

Figure 3: Results of a state of the art coreference system trained on silver chains.

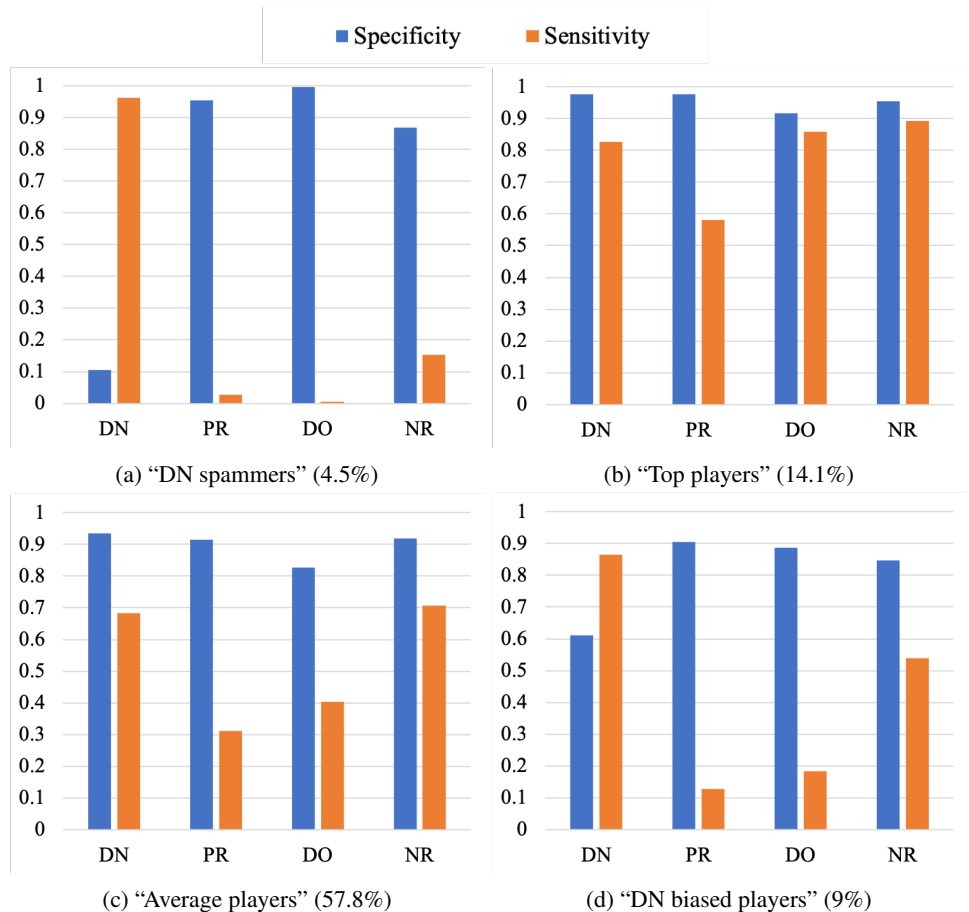

Figure 4: Examples of inferred user communities from the *Phrase Detectives 2* corpus. The horizontal line refers to the average ability for DN - discourse new interpretations, PR - property (predicative NPs), DO - discourse old and NR - non referring (expletives). Each community is given an intuitive name and is presented together with its prevalence.

Figure 4 presents the community profiles the players from the aforementioned lists belong to. Interestingly, all the spammers were assigned to the community from Figure 4a. The model assigns almost full probability mass to the average specificity of this community and negligible mass to the sensitivity for all classes with the exception of DN, where this mass assignment is reversed. For that to objectively be the case, the players under this community would have to (almost) always

|                | Bluebird         | RTE              | Valence          | Temp              |
|----------------|------------------|------------------|------------------|-------------------|
| Majority vote  | 75.93 (82/108)   | 91.88 (735/800)  | 80.00 (80/100)   | 93.94 (434/462)   |
| iBCC           | 89.81 (97/108)   | 92.88 (743/800)  | 85.00 (85/100)   | 94.35 (436/462)   |
| cBCC           | 88.89 (96/108)   | 93.12 (745/800)  | 88.00 (88/100)   | 94.37 (436/462)   |
| hcBCC          | 88.89 (96/108)   | 93.12 (745/800)  | 89.00 (89/100)   | 94.37 (436/462)   |
| MPA            | 88.89 (96/108)   | 93.00 (744/800)  | 86.00 (86/100)   | 94.16 (435/462)   |
| COMMUNITYMPA   | 89.81 (97/108)   | 93.25 (746/800)  | 85.00 (85/100)   | 94.16 (435/462)   |

Table 1: Accuracy (correctly adjudicated items / total number of items) results on traditional crowd-sourcing datasets. The results of the first 4 methods are as reported in (Moreno et al., 2015).

choose the DN interpretation, an aspect confirmed by an inspection of their annotations. The annotators from the list of honest players, have more diverse ability profiles. Most of them belong to the pool of "Top players" presented in Figure 4b. The players from this community have a solid understanding of anaphoric annotation and generally a large workload; for example the 10 people from the supplied list that the model assigned to this community have provided 35% of the entire corpus annotations. Figure 4c shows the profile of the average players, named so based on the large prevalence of this community in the population of annotators. Looking at their sensitivity, the estimates confirm the general intuition that PR cases (predicative NPs) are the hardest to spot, mostly because they can easily be confused with DN (discourse new); indefinite NPs (e.g., a policeman) are the most common type of mention in both classes. Expletives (NR) and the introduction of new entities into the discourse (DN) are the easiest to understand by most people. Another intuitive result is that discourse old mentions are more difficult compared with discourse new ones; the former requires the identification of the most recent antecedent, which could go wrong for various reasons (e.g., ambiguity, negligence). Finally, we could also find in the supplied list players who are biased towards DN, but make the effort occasionally to supply other interpretations as well (see Figure 4d).

## 3.5 TRADITIONAL CROWDSOURCING TASKS

Both COMMUNITYMPA and its unpooled counterpart MPA can be applied to traditional crowdsourcing datasets where the set of classes the coders can choose from is the same across the annotated items. Under the modeling framework described in this work (Section 2), for this type of data, the labels coincide with the classes they belong to. We compare the aforementioned models against the methods from Moreno et al. (2015) which include a majority vote baseline, the iBCC model of Kim & Ghahramani (2012) which is a Bayesian version of the model presented in the seminar work of Dawid & Skene (1979), and two nonparametric community models: the cBCC model where annotators are assigned into communities and annotate according to the profile of the community they belong to, and the hBCC model which assumes both annotator and community level structures. The latter model is a nonparametric extention of the Dawid & Skene (1979) model, similarly how COMMUNITYMPA is to MPA.

The evaluation was conducted on 4 datasets commonly used in the crowdsourcing literature (Moreno et al., 2015; Raykar & Yu, 2012; Snow et al., 2008; Hovy et al., 2013; Paun et al., 2018a). Table 1 presents the accuracy results, indicating on par performance between the probabilistic models and a slight advantage of these models against the majority vote baseline. The simple majority vote baseline implicitly assumes an equal expertise among annotators, a well known shortcoming, previous studies reporting more significant differences in performance to the probabilistic models of annotation, on larger datasets (Venanzi et al., 2014; Paun et al., 2018a).

## 4 RELATED WORK

To our knowledge, the model presented in this work and its unpooled counterpart from Paun et al. (2018b) are the first models designed for anaphoric annotation. Both models however draw inspiration from the mention-pair model of coreference (Soon et al., 2001; Hoste, 2016) and from traditional models of annotation.

The model we introduced in this work functions as a component in a rather abstract mention-pair framework: the mention pairs are inferred in an unsupervised fashion from the crowd annotations and the clustering to coreference chains is done by simply following the unique link structure from the inferred pairs.

Traditional models of annotation assume the set of classes the annotators can choose from is fixed across the annotated items, an aspect not appropriate for an anaphoric annotation task. However, the model we introduced is after all a model of annotation, and thus shares many of the characteristics found in this area of research. Back in the late 70's, in a seminal work, Dawid & Skene (1979) introduced a model of annotation that would take into account in its inference for the correct interpretations the ability of the annotators. The model has found wide application and inspired the state of the art through the years (Smyth et al., 1995; Albert & Dodd, 2004; Whitehill et al., 2009; Raykar et al., 2010; Kim & Ghahramani, 2012; Hovy et al., 2013; Simpson et al., 2013; Passonneau & Carpenter, 2014; Felt et al., 2014; 2015; Kamar et al., 2015; Venanzi et al., 2014; Moreno et al., 2015; Nguyen et al., 2017; Paun et al., 2018a, inter alia). Two of the models proposed in recent years assume a partially pooled structure, having the annotators clustered into communities. This type of structure uses information about the communities to improve the estimates of the individuals by regularizing towards the community mean. One of the models assumes a fixed number of clusters (Venanzi et al., 2014), while the other takes a nonparametric approach letting the number of clusters grow with the data (Moreno et al., 2015). Both of these models are community extensions of the Dawid & Skene (1979) model. The model we introduced in this work also assumes a partially pooled structure, but it was built as an extension of the MPA model for anaphoric annotations (Paun et al., 2018b). Compared to the nonparametric community model of Moreno et al. (2015), the closest of the two community models to ours, besides the differences in the data these models were designed for, there are also differences in the parameterization and inference. Moreno et al. (2015) use a Chinese restaurant process to allow for a latent number of communities and Gibbs sampling for inference, whereas we use a stick breaking process with variational inference, aiming for conjugacy throughout our parameterization. Since our model can also be used on traditional crowdsourcing datasets, we showed back in Section 3.5 that we get comparable performance to the state of the art which included the nonparametric model of Moreno et al. (2015).

## 5 CONCLUSIONS

The development of more powerful and versatile coreference resolution systems relies on the availability of larger and linguistically richer datasets, and crowdsourcing has been identified as a viable alternative to expert annotation, offering comparable quality at a fraction of the costs and larger scalability. Although the study of models of annotation, necessary to adjudicating crowd labels, received much attention over the years, it was mostly aimed at standard classification tasks where the set of classes the coders can choose from is the same across the annotated items. The literature on models of anaphoric annotation is scarce, the only previous effort in this direction being the recently introduced mention pair model of Paun et al. (2018b).

In this work we extended the unpooled model proposed by Paun et al. (2018b) with hierarchical communities of annotators, using information about the community to improve the estimates of the individuals. The partially pooled extension is nonparametric, letting the number of communities grow with the data, flexibility we achieved with the help of a Dirichlet process mixture based on a stick-breaking representation of the underlying Dirichlet process. The hierarchical structure offers a better resilience to sparsity, making the model a better fit (compared to its unpooled counterpart) for a larger number of crowdsourcing setups. We demonstrated this across a number of coreference resolution related tasks, in various levels of sparsity, assessing the accuracy of the inferred mention pairs, the quality of the post-hoc constructed silver chains, and the viability of using silver chains as an alternative to expert-annotated chains when training a state of the art coreference system. We also included a discussion of the inferred community profiles.

The model, although developed for anaphoric annotation, is also flexible enough to be used in traditional crowdsourcing setups where the set of classes the coders can choose from is the same across the annotated items. We showed, in this context, the model is on par with the state of the art.

The paper also includes guidance for the estimation of the parameters using variational inference (see appendix) and is accompanied by the code implementing the proposed model.

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

## A  PARAMETER ESTIMATION DETAILS

In this appendix we first list all the formulas for the variational parameters that are required to fully reimplement the model. This is followed by a section that provides additional details on how to derive some of the more challenging formulas.

### A.1  NOTES ON THE UPDATE FORMULAS OF THE VARIATIONAL PARAMETERS

The derivations for the formulas required to compute the variational parameters follow a straightforward procedure: 1) compute the complete conditionals of each parameter, 2) set the variational distributions to the same exponential family form as their corresponding complete conditionals, and 3) use the fact that the natural parameters of the variational distribution are equal to the expected value (under the variational distribution) of the natural parameters of the complete conditionals.

#### A.1.1  MENTION-LEVEL PARAMETERS (AND PRIORS)

It is straightforward to show the complete conditional for the class-level true label likelihood $p(\pi_h|...)$ is a Beta distribution. This gives us $q(\pi_h|\lambda_h, \eta_h) = \mathsf{Beta}(\lambda_h, \eta_h)$, where:

$$\lambda_h = a_0 + \sum_{i,m}^{I,M_i} \mathsf{I}(z_{i,m} = h)\mathsf{E}_q[\mathsf{I}(c_{i,m} = 1)] \tag{1}$$

$$\eta_h = a_1 + \sum_{i,m}^{I,M_i} \mathsf{I}(z_{i,m} = h)\mathsf{E}_q[\mathsf{I}(c_{i,m} = 0)] \tag{2}$$

The conditional for the true label indicators $p(c_{i,m}|...)$ is Bernoulli distributed. This gives us $q(c_{i,m}|\phi_{i,m}) = \mathsf{Bernoulli}(\phi_{i,m})$, where $\zeta_{i,m} = 1 - \phi_{i,m}$ and:

$$\log \phi_{i,m} = \mathsf{E}_q[\log \pi_{z_{i,m}}] + \sum_{n=1}^{N_{i,m}} \log \Upsilon(\delta_{i,m,n}) + \mathsf{E}_q[\alpha_{jj[i,m,n],z_{i,m}}]y_{i,m,n} -$$
$$0.5(\mathsf{E}_q[\alpha_{jj[i,m,n],z_{i,m}}] + \delta_{i,m,n}) - \Lambda(\delta_{i,m,n})(\mathsf{E}_q[\alpha_{jj[i,m,n],z_{i,m}}^2] - \delta_{i,m,n}^2) + const. \tag{3}$$

$$
\log \zeta_{i,m} = \mathsf{E}_q[\log(1 - \pi_{z_{i,m}})] + \sum_{n=1}^{N_{i,m}} \log \Upsilon(\rho_{i,m,n}) - \mathsf{E}_q[\beta_{jj[i,m,n],z_{i,m}}]y_{i,m,n} +
$$
$$
0.5(\mathsf{E}_q[\beta_{jj[i,m,n],z_{i,m}}] - \rho_{i,m,n}) - \Lambda(\rho_{i,m,n})(\mathsf{E}_q[\beta^2_{jj[i,m,n],z_{i,m}}] - \rho^2_{i,m,n}) + const.
$$
$$(4)$$

The solution for the variational parameters introduced by the Jaakkola & Jordan (2000) bound can easily be found by taking partial derivatives of the variational objective function (the evidence lower-bound):

$$
\delta^2_{i,m,n} = \mathsf{E}_q[\alpha^2_{jj[i,m,n],z_{i,m}}]
$$
$$(5)$$

$$
\rho^2_{i,m,n} = \mathsf{E}_q[\beta^2_{jj[i,m,n],z_{i,m}}]
$$
$$(6)$$

### A.1.2 ANNOTATOR-LEVEL PARAMETERS

The complete conditional of an annotator's sensitivity $p(\alpha_{j,h}|...)$ is a Gaussian distribution. Therefore, we have $q(\alpha_{j,h}|\gamma_{j,h}, \mu_{j,h}) = \mathsf{Normal}(\gamma_{j,h}, \mu_{j,h})$, where:

$$
\mu^{-1}_{j,h} = \sum_{r=1}^{C} \mathsf{E}_q[\mathsf{I}(x_j = r)]\mathsf{E}_q[\sigma^{-1}_{r,h}] +
$$
$$
2 \sum_{i,m,n} \mathsf{I}(jj[i,m,n] = j)\mathsf{I}(z_{i,m} = h)\mathsf{E}_q[\mathsf{I}(c_{i,m} = 1)]\lambda(\delta_{i,m,n})
$$
$$(7)$$

$$
\gamma_{j,h} = \mu_{j,h}(\sum_{r=1}^{C} \mathsf{E}_q[\mathsf{I}(x_j = r)]\mathsf{E}_q[\tau_{r,h}]\mathsf{E}_q[\sigma^{-1}_{r,h}] +
$$
$$
\sum_{i,m,n} \mathsf{I}(jj[i,m,n] = j)\mathsf{I}(z_{i,m} = h)\mathsf{E}_q[\mathsf{I}(c_{i,m} = 1)](y_{i,m,n} - 0.5))
$$
$$(8)$$

Similarly, the complete conditional of an annotator's specificity $p(\beta_{j,h}|...)$ is also a Gaussian distribution. We have $q(\beta_{j,h}|\theta_{j,h}, \epsilon_{j,h}) = \mathsf{Normal}(\theta_{j,h}, \epsilon_{j,h})$, where:

$$
\epsilon^{-1}_{j,h} = \sum_{r=1}^{C} \mathsf{E}_q[\mathsf{I}(x_j = r)]\mathsf{E}_q[\Omega^{-1}_{r,h}] +
$$
$$
2 \sum_{i,m,n} \mathsf{I}(jj[i,m,n] = j)\mathsf{I}(z_{i,m} = h)\mathsf{E}_q[\mathsf{I}(c_{i,m} = 0)]\lambda(\rho_{i,m,n})
$$
$$(9)$$

$$
\theta_{j,h} = \epsilon_{j,h}(\sum_{r=1}^{C} \mathsf{E}_q[\mathsf{I}(x_j = r)]\mathsf{E}_q[\nu_{r,h}]\mathsf{E}_q[\Omega^{-1}_{r,h}] +
$$
$$
\sum_{i,m,n} \mathsf{I}(jj[i,m,n] = j)\mathsf{I}(z_{i,m} = h)\mathsf{E}_q[\mathsf{I}(c_{i,m} = 0)](0.5 - y_{i,m,n}))
$$
$$(10)$$

The community indicator $p(x_j|...)$ has its complete conditional categorically distributed. This gives us $q(x_j|\kappa_j) = \mathsf{Categorical}(\kappa_j)$, where:

$$\log \kappa_{j,r} = \mathsf{E}_q[\log \Delta_r] + \sum_{r'=1}^{r-1} \mathsf{E}_q[\log(1 - \Delta_{r'})]+$$

$$\sum_{h=1}^{K} (-0.5) \left( \mathsf{E}_q[\log \sigma_{r,h}] + \mathsf{E}_q[\sigma_{r,h}^{-1}]\mathsf{E}_q[(\alpha_{j,h} - \tau_{r,h})^2] \right) + \tag{11}$$

$$(-0.5) \left( \mathsf{E}_q[\log \Omega_{r,h}] + \mathsf{E}_q[\Omega_{r,h}^{-1}]\mathsf{E}_q[(\beta_{j,h} - \nu_{r,h})^2] \right) + const.$$

### A.1.3 Community-level parameters (and priors)

The complete conditional of the class-level mean sensitivity of a community $p(\tau_{r,h}|...)$ is a Gaussian. That means we have $q(\tau_{r,h}|\tau_{r,h}^{\mu}, \tau_{r,h}^{\sigma}) = \mathsf{Normal}(\tau_{r,h}^{\mu}, \tau_{r,h}^{\sigma})$, where:

$$\tau_{r,h}^{\sigma} = \left( d_1^{-1} + \mathsf{E}_q[\sigma_{r,h}^{-1}] \sum_{j=1}^{J} \mathsf{E}_q[\mathsf{I}(x_j = r)] \right)^{-1} \tag{12}$$

$$\tau_{r,h}^{\mu} = \tau_{r,h}^{\sigma} \left( d_0/d_1 + \mathsf{E}_q[\sigma_{r,h}^{-1}] \sum_{j=1}^{J} \mathsf{E}_q[\mathsf{I}(x_j = r)]\mathsf{E}_q[\alpha_{j,h}] \right) \tag{13}$$

The conditional for the class-level variance of the sensitivity of a community $p(\sigma_{r,h}|...)$ is an Inverse-Gamma distribution. That leads to $q(\sigma_{r,h}|\sigma_{r,h}^{\alpha}, \sigma_{r,h}^{\beta}) = \mathsf{Inverse-Gamma}(\sigma_{r,h}^{\alpha}, \sigma_{r,h}^{\beta})$, where:

$$\sigma_{r,h}^{\alpha} = e_0 + 0.5 \sum_{j=1}^{J} \mathsf{E}_q[\mathsf{I}(x_j = r)] \tag{14}$$

$$\sigma_{r,h}^{\beta} = e_1 + 0.5 \sum_{j=1}^{J} \mathsf{E}_q[\mathsf{I}(x_j = r)]\mathsf{E}_q[(\alpha_{j,h} - \tau_{r,h})^2] \tag{15}$$

A similar discussion applies to the specificity-related parameters of a community. The conditional of the mean specificity $p(\nu_{r,h}|...)$ is a Gaussian, leading to $q(\nu_{r,h}|\nu_{r,h}^{\mu}, \nu_{r,h}^{\sigma}) = \mathsf{Normal}(\nu_{r,h}^{\mu}, \nu_{r,h}^{\sigma})$, where:

$$\nu_{r,h}^{\sigma} = \left( t_1^{-1} + \mathsf{E}_q[\Omega_{r,h}^{-1}] \sum_{j=1}^{J} \mathsf{E}_q[\mathsf{I}(x_j = r)] \right)^{-1} \tag{16}$$

$$\nu_{r,h}^{\mu} = \nu_{r,h}^{\sigma} \left( d_0/d_1 + \mathsf{E}_q[\Omega_{r,h}^{-1}] \sum_{j=1}^{J} \mathsf{E}_q[\mathsf{I}(x_j = r)]\mathsf{E}_q[\beta_{j,h}] \right) \tag{17}$$

The complete conditional for the variance of the specificity $p(\Omega_{r,h}|...)$ is an Inverse-Gamma distribution, giving us $q(\Omega_{r,h}|\Omega_{r,h}^{\alpha}, \Omega_{r,h}^{\beta}) = \mathsf{Inverse-Gamma}(\Omega_{r,h}^{\alpha}, \Omega_{r,h}^{\beta})$, where:

$$\Omega_{r,h}^{\alpha} = u_0 + 0.5 \sum_{j=1}^{J} \mathsf{E}_q[\mathsf{I}(x_j = r)] \tag{18}$$

$$\Omega_{r,h}^{\beta} = u_1 + 0.5 \sum_{j=1}^{J} \mathsf{E}_q[\mathsf{I}(x_j = r)]\mathsf{E}_q[(\beta_{j,h} - \nu_{r,h})^2] \tag{19}$$

| Var. | Summary | ‖ Var. | Summary |
|------|---------|---------|---------|
| $c_{i,m}$ | $q(c_{i,m}) = \text{Bernoulli}(\phi_{i,m})$ $\mathsf{E}_q[\mathsf{I}(c_{i,m}=1)] = \phi_{i,m}$ $\mathsf{E}_q[\mathsf{I}(c_{i,m}=0)] = \zeta_{i,m}$ | $\nu_{r,h}$ | $q(\nu_{r,h}) = \text{Normal}(\nu_{r,h}^\mu, \nu_{r,h}^\sigma)$ $\mathsf{E}_q[\nu_{r,h}] = \nu_{r,h}^\mu$ $\mathsf{E}_q[\nu_{r,h}^2] = (\nu_{r,h}^\mu)^2 + \nu_{r,h}^\sigma$ |
| $x_j$ | $q(x_j) = \text{Categorical}(\kappa_j)$ $\mathsf{E}_q[\mathsf{I}(x_j=r)] = \kappa_{j,r}$ $\mathsf{E}_q[\mathsf{I}(x_j>r)] = \sum_{r'=r+1}^{C} \kappa_{j,r'}$ | $\Omega_{r,h}$ | $q(\Omega_{r,h}) = \text{Inverse}-\text{Gamma}(\Omega_{r,h}^\alpha, \Omega_{r,h}^\beta)$ $\mathsf{E}_q[\log \Omega_{r,h}] = \log \Omega_{r,h}^\beta - \Psi(\Omega_{r,h}^\alpha)$ $\mathsf{E}_q[\Omega_{r,h}^{-1}] = \frac{\Omega_{r,h}^\alpha}{\Omega_{r,h}^\beta}$ |
| $\alpha_{j,h}$ | $q(\alpha_{j,h}) = \text{Normal}(\gamma_{j,h}, \mu_{j,h})$ $\mathsf{E}_q[\alpha_{j,h}] = \gamma_{j,h}$ $\mathsf{E}_q[\alpha_{j,h}^2] = \gamma_{j,h}^2 + \mu_{j,h}$ | $\Delta_r$ | $q(\Delta_r) = \text{Beta}(f_r, g_r)$ $\mathsf{E}_q[\log \Delta_r] = \Psi(f_r) - \Psi(f_r+g_r)$ $\mathsf{E}_q[\log(1-\Delta_r)] = \Psi(g_r) - \Psi(f_r+g_r)$ |
| $\beta_{j,h}$ | $q(\beta_{j,h}) = \text{Normal}(\theta_{j,h}, \epsilon_{j,h})$ $\mathsf{E}_q[\beta_{j,h}] = \theta_{j,h}$ $\mathsf{E}_q[\beta_{j,h}^2] = \theta_{j,h}^2 + \epsilon_{j,h}$ | $\pi_h$ | $q(\pi_h)] = \text{Beta}(\lambda_h, \eta_h)$ $\mathsf{E}_q[\log \pi_h] = \Psi(\lambda_h) - \Psi(\lambda_h+\eta_h)$ $\mathsf{E}_q[\log(1-\pi_h)] = \Psi(\eta_h) - \Psi(\lambda_h+\eta_h)$ |
| $\tau_{r,h}$ | $q(\tau_{r,h}) = \text{Normal}(\tau_{r,h}^\mu, \tau_{r,h}^\sigma)$ $\mathsf{E}_q[\tau_{r,h}] = \tau_{r,h}^\mu$ $\mathsf{E}_q[\tau_{r,h}^2] = (\tau_{r,h}^\mu)^2 + \tau_{r,h}^\sigma$ | $b$ | $q(b) = \text{Gamma}(b^\alpha, b^\beta)$ $\mathsf{E}_q[b] = \frac{b^\alpha}{b^\beta}$ $\mathsf{E}_q[\log b] = \Psi(b^\alpha) - \log b^\beta$ |
| $\sigma_{r,h}$ | $q(\sigma_{r,h}) = \text{Inverse}-\text{Gamma}(\sigma_{r,h}^\alpha, \sigma_{r,h}^\beta)$ $\mathsf{E}_q[\log \sigma_{r,h}] = \log \sigma_{r,h}^\beta - \Psi(\sigma_{r,h}^\alpha)$ $\mathsf{E}_q[\sigma_{r,h}^{-1}] = \frac{\sigma_{r,h}^\alpha}{\sigma_{r,h}^\beta}$ | | |

Table 2: A summary of the variational distributions together with relevant expectations.

The conditional for the stick proportions $p(\Delta_r|...)$ is a Beta distribution. The corresponding variational distribution has the same form $q(\Delta_r|f_r, g_r) = \text{Beta}(f_r, g_r)$, where:

$$f_r = 1 + \sum_{j=1}^{J} \mathsf{E}_q[\mathsf{I}(x_j = r)] \tag{20}$$

$$g_r = b + \sum_{j=1}^{J} \mathsf{E}_q[\mathsf{I}(x_j > r)] \tag{21}$$

Finally, the conditional for the scale $p(b|...)$ is a Gamma distribution, which means $q(b|b^\alpha, b^\beta) = \text{Gamma}(b^\alpha, b^\beta)$, where:

$$b^\alpha = s_0 + C - 1 \tag{22}$$

$$b^\beta = s_1 - \sum_{r=1}^{C-1} \mathsf{E}_q[\log(1-\Delta_r)] \tag{23}$$

### A.1.4 IMPLEMENTATION NOTES

The parameters are updated in turn until the variational objective function (the evidence lowerbound) plateaus. The algorithm is often referred to as Coordinate Ascent (mean field) Variational Inference (CAVI) (Blei et al., 2017). Table 2 summarizes the variational distributions together with the relevant expectations required for the implementation.

For initialization we used the annotator profiles inferred by MPA together with random community assignments. MPA was run with the default configuration: uniform priors, with the sensitivity

and specificity of the annotators receiving a bit of extra mass at initialization to address the label switching problem. This problem, specific to mixture models which are at the core of many models of annotation, is caused by the likelihood's invariance under the permutation of the labels, making the models nonidentifiable. Regarding the choice of hyper-parameters, we used either uniform or weakly informative priors that inform the scale of the results, but are not otherwise sensitive $(a_0 = a_1 = s_0 = s_1 = d_1 = t_1 = e_1 = u_1 = 1, d_0 = t_0 = 0, e_0 = u_0 = 5)$. The truncation level was set to $C = 10$ in all of our experiments.

## A.2 FURTHER DETAILS

This subsection provides additional parameter estimation details for the readers interested in deriving the update formulas presented above. As stated back in Section 2.2, we parameterized the model with conjugacy in mind, and where that was not directly possible, i.e., in those cases involving the standard logistic function, we addressed the problem using the bound from Jaakkola & Jordan (2000). We include below additional guidance for these cases. Focusing on the sensitivity side of the derivations (specificity is analogous) we have:

$$
\begin{aligned}
p(y_{i,m,n}|\alpha_{jj[i,m,n],z_{i,m}}) &= \mathsf{Bernoulli}(\Upsilon(\alpha_{jj[i,m,n],z_{i,m}})) \\
&= \Upsilon(\alpha_{jj[i,m,n],z_{i,m}})^{y_{i,m,n}}(1 - \Upsilon(\alpha_{jj[i,m,n],z_{i,m}}))^{1-y_{i,m,n}} \\
&= \exp\{\alpha_{jj[i,m,n],z_{i,m}}y_{i,m,n}\}\Upsilon(-\alpha_{jj[i,m,n],z_{i,m}})
\end{aligned}
\tag{24}
$$

The bound from Jaakkola & Jordan (2000) gives us:

$$
\begin{aligned}
\Upsilon(\alpha_{jj[i,m,n],z_{i,m}}) \geq &\Upsilon(\delta_{i,m,n})\exp\{(\alpha_{jj[i,m,n],z_{i,m}} - \delta_{i,m,n})/2- \\
&\Lambda(\delta_{i,m,n})(\alpha_{jj[i,m,n],z_{i,m}}^2 - \delta_{i,m,n}^2)\}
\end{aligned}
\tag{25}
$$

where
$$
\Lambda(\delta_{i,m,n}) = 0.5\delta_{i,m,n}^{-1}(\Upsilon(\delta_{i,m,n}) - 0.5)
$$

Using the bound back in Equation 24 leads us to:

$$
\begin{aligned}
p(y_{i,m,n}|\alpha_{jj[i,m,n],z_{i,m}}) \geq \Upsilon(\delta_{i,m,n})\exp\{\alpha_{jj[i,m,n],z_{i,m}}y_{i,m,n}- \\
(\alpha_{jj[i,m,n],z_{i,m}} + \delta_{i,m,n})/2 - \Lambda(\delta_{i,m,n})(\alpha_{jj[i,m,n],z_{i,m}}^2 - \delta_{i,m,n}^2)\}
\end{aligned}
\tag{26}
$$

Drawing annotations based on the sensitivity of the annotators comes up in two complete conditionals, one related to the sensitivity itself, and the other in the positive outcome of the true label indicators. The first conditional is:

$$
p(\alpha_{j,h}|...) \propto p(\alpha_{j,h}|\tau_{x_j,h}, \sigma_{x_j,h})\prod_{i,m,n}^{I,M_i,N_{i,m}}p(y_{i,m,n}|\alpha_{j,h})^{\mathsf{I}(jj[i,m,n]=j)\mathsf{I}(z_{i,m}=h)\mathsf{I}(c_{i,m}=1)}
\tag{27}
$$

The first term can be expressed as:

$$
p(\alpha_{j,h}|\tau_{x_j,h}, \sigma_{x_j,h}) \propto \exp\left\{-0.5\sum_{r=0}^{\infty}\mathsf{I}(x_j = r)\sigma_{r,h}^{-1}(\alpha_{j,h}^2 - 2\alpha_{j,h}\tau_{r,h})\right\}
\tag{28}
$$

The second term is:

$$
\begin{aligned}
p(y_{i,m,n}|\alpha_{j,h}) &= e^{\alpha_{j,h}y_{i,m,n}}\Upsilon(-\alpha_{j,h}) \\
&\geq \Upsilon(\delta_{i,m,n})\exp\left\{\alpha_{j,h}y_{i,m,n} - (\alpha_{j,h} + \delta_{i,m,n})/2 - \Lambda(\delta_{i,m,n})(\alpha_{j,h}^2 - \delta_{i,m,n}^2)\right\} \\
&\propto_{\alpha_{j,h}} \exp\left\{\alpha_{j,h}y_{i,m,n} - 0.5\alpha_{j,h} - \Lambda(\delta_{i,m,n})\alpha_{j,h}^2\right\}
\end{aligned}
\tag{29}
$$

Going back to Equation 27 we have:

$$p(\alpha_{j,h}|...) \propto \exp\left\{-0.5 \sum_{r=1}^{\infty} \mathsf{I}(x_j = r)\sigma_{r,h}^{-1}(\alpha_{j,h}^2 - 2\alpha_{j,h}\tau_{r,h})\right\} \times$$

$$\times \prod_{i,m,n} \left(\exp\left\{\alpha_{j,h}y_{i,m,n} - 0.5\alpha_{j,h} - \Lambda(\delta_{i,m,n})\alpha_{j,h}^2\right\}\right) \underbrace{\mathsf{I}(jj[i,m,n] = j)\mathsf{I}(z_{i,m} = h)\mathsf{I}(c_{i,m} = 1)}_{\text{shorten as } \mathsf{I}(\dots)}$$

$$\propto \exp\{-0.5 \sum_{r=1}^{\infty} \mathsf{I}(x_j = r)\sigma_{r,h}^{-1}(\alpha_{j,h}^2 - 2\alpha_{j,h}\tau_{r,h}) +$$

$$+ \sum_{i,m,n}^{I,M_i,N_{i,m}} \mathsf{I}(...)(\alpha_{j,h}y_{i,m,n} - 0.5\alpha_{j,h} - \Lambda(\delta_{i,m,n})\alpha_{j,h}^2)\}$$

$$\propto \exp\{-0.5[\alpha_{j,h}^2(\underbrace{\sum_{r=1}^{\infty}\mathsf{I}(x_j=r)\sigma_{r,h}^{-1} + 2\sum_{i,m,n}\mathsf{I}(...)\Lambda(\delta_{i,m,n}))}_{\text{let's call this } a} -$$

$$- 2\alpha_{j,h}(\underbrace{\sum_{r=1}^{\infty}\mathsf{I}(x_j=r)\sigma_{r,h}^{-1}\tau_{r,h} + \sum_{i,m,n}\mathsf{I}(...)(y_{i,m,n} - 0.5))}_{\text{let's call this } b})]\}$$

$$\propto \exp\left\{-0.5a\left(\alpha_{j,h}^2 - 2\alpha_{j,h}\frac{b}{a}\right)\right\}$$

$$\propto \exp\left\{-0.5\left(\frac{1}{a}\right)^{-1}\left(\alpha_{j,h} - \frac{b}{a}\right)^2\right\}$$

$$= \mathsf{Normal}(b/a, 1/a)$$

$$\tag{30}$$

We have just showed the complete conditional of the annotator's sensitivity $p(\alpha_{j,h}|...)$ is a Gaussian distribution. The corresponding variational distribution will take the same form, i.e., $q(\alpha_{j,h}|\gamma_{j,h}, \mu_{j,h}) = \mathsf{Normal}(\gamma_{j,h}, \mu_{j,h})$. Using the aforementioned connection between the natural parameters of the two distributions leads us to the formulas presented back in Section A.1.2.

The other complete conditional which involves the bounded logistic function is the true label indicator. Again, focusing on the sensitivity side of the dependencies, we have:

$$p(c_{i,m} = 1|...) \propto p(c_{i,m} = 1|\pi_{z_{i,m}}) \prod_{n=1}^{N_{i,m}} p(y_{i,m,n}|\alpha_{jj[i,m,n],z_{i,m}})$$

$$\propto \exp\left\{\log \pi_{z_{i,m}} + \sum_{n=1}^{N_{i,m}} \log p(y_{i,m,n}|\alpha_{jj[i,m,n],z_{i,m}})\right\}$$

$$\tag{31}$$

After expanding $p(c_{i,m} = 0|...)$ in a similar manner, it can be seen the conditional for the true label indicators, $p(c_{i,m}|...)$, is Bernoulli distributed. This means the corresponding variational distribution takes the same form, i.e., $q(c_{i,m}|\phi_{i,m}) = \mathsf{Bernoulli}(\phi_{i,m})$. To get the update formula of the variational parameter we simply take the necessary expectations:

$$\log \phi_{i,m} = \mathsf{E}_q[\log \pi_{z_{i,m}}] + \sum_{n=1}^{N_{i,m}} \mathsf{E}_q[\log p(y_{i,m,n}|\alpha_{jj[i,m,n],z_{i,m}})] + const. \tag{32}$$

The first expectation is trivial, while for the second one we need to apply the bound from Equation 25. A few further derivations should result in the update formulas presented back in Section A.1.1.

The Jaakkola & Jordan (2000) bound introduces an additional variational parameter (see Equation 25) which can be found by taking partial derivatives of the variational objective function (the ELBO). For example, the terms dependent on $\delta_{i,m,n}$ are:

$$\mathcal{L}_{\delta_{i,m,n}} = \phi_{i,m} \left\{ \log \Upsilon(\delta_{i,m,n}) - 0.5\delta_{i,m,n} - \Lambda(\delta_{i,m,n}) \left( \mathsf{E}_q[\alpha^2_{jj[i,m,n],z_{i,m}}] - \delta^2_{i,m,n} \right) \right\} \quad (33)$$

Deriving the ELBO with respect to $\delta_{i,m,n}$ leads to:

$$(\mathcal{L}_{\delta_{i,m,n}})' = \phi_{i,m}\{(\Lambda(\delta_{i,m,n}))'(\mathsf{E}_q[\alpha^2_{jj[i,m,n],z_{i,m}}] - \delta^2_{i,m,n})\} \quad (34)$$

Setting the above equation to 0 and solving for $\delta_{i,m,n}$ gives the result stated in Section A.1.1.

Computing the other complete conditionals and deriving the rest of the update formulas presented in the paper should be straightforward due to conjugacy; simply follow the three-steps procedure outlined in the beginning of this appendix and put in practice above. The stick breaking representation of the Dirichlet process nicely complies with conjugacy as well; the notes from Blei & Jordan (2006) can be a useful read. Finally, following all of the steps up to this point, should also make it straightforward to expand the variational objective function (ELBO) used to assess the convergence of the coordinate ascent algorithm.

## B    DETAILED EVALUATION RESULTS

This Appendix includes a detailed view of the evaluation results presented in the paper. Specifically, Table 3 shows more details for the inferred mention pairs evaluated back in Section 3.1, Table 4 presents a deeper view into the quality of the silver chains discussed in Section 3.2, while Tables 5 and 6 show an extended view of the performance obtained by a state of the art coreference system when trained on silver chains as it was presented back in Section 3.3.

|  | # | Acc. | Property | | | Non referring | | | Discourse new | | | Discourse old | | |
|---|---|---|---|---|---|---|---|---|---|---|---|---|---|---|
|  | | | P | R | F1 | P | R | F1 | P | R | F1 | P | R | F1 |
| MPA | 5 | 86.2 | 100.0 | 0.7 | 1.4 | 92.3 | 24.5 | 38.7 | 83.9 | 98.7 | 90.7 | 91.4 | 73.7 | 81.6 |
| MPA | 10 | 88.9 | 100.0 | 6.2 | 11.7 | 100.0 | 57.1 | 72.7 | 88.5 | 97.5 | 92.8 | 89.5 | 81.6 | 85.4 |
| MPA | 20 | 90.5 | 69.4 | 17.2 | 27.6 | 100.0 | 69.4 | 81.9 | 91.7 | 96.7 | 94.1 | 88.6 | 86.0 | 87.3 |
| MPA | 40 | 91.6 | 74.1 | 29.7 | 42.4 | 97.6 | 81.6 | 88.9 | 93.2 | 96.7 | 94.9 | 89.3 | 88.0 | 88.6 |
| MPA | all | 92.2 | 64.0 | 72.4 | 68.0 | 94.1 | 98.0 | 96.0 | 94.5 | 96.0 | 95.3 | 90.4 | 87.3 | 88.8 |
| CMPA | 5 | 88.6 | 76.7 | 15.9 | 26.3 | 97.1 | 69.4 | 81.0 | 87.1 | 98.1 | 92.3 | 91.7 | 78.9 | 84.8 |
| CMPA | 10 | 91.3 | 72.2 | 39.3 | 50.9 | 92.3 | 98.0 | 95.1 | 92.9 | 96.3 | 94.6 | 89.3 | 86.7 | 88.0 |
| CMPA | 20 | 91.7 | 64.8 | 46.9 | 54.4 | 94.1 | 98.0 | 96.0 | 93.9 | 96.1 | 95.0 | 89.3 | 87.6 | 88.4 |
| CMPA | 40 | 92.0 | 66.4 | 58.6 | 62.3 | 94.0 | 95.9 | 95.0 | 94.4 | 95.9 | 95.1 | 89.7 | 88.2 | 88.9 |
| CMPA | all | 92.2 | 64.0 | 72.4 | 68.0 | 94.1 | 98.0 | 96.0 | 94.5 | 96.0 | 95.3 | 90.6 | 87.4 | 89.0 |
| MV | - | 82.9 | 53.9 | 9.7 | 16.4 | 97.2 | 71.4 | 82.4 | 79.1 | 99.0 | 87.9 | 94.5 | 62.8 | 75.4 |

Table 3: Detailed evaluation results of the inferred mention pairs matched against expert annotations. The "#" column shows the maximum user workload where the "all" configuration uses the dataset as it is (does not alter the player workloads). CMPA is shorthand for COMMUNITYMPA and MV for majority voting.

| | Method | | MUC | | | BCUB | | | CEAFE | | | Avg. F1 |
|---|---|---|---|---|---|---|---|---|---|---|---|---|
| | | # | P | R | F1 | P | R | F1 | P | R | F1 | |
| Singletons included | MPA | 5 | 93.3 | 76.3 | 84.0 | 94.0 | 83.9 | 88.7 | 81.9 | 95.2 | 88.1 | 86.9 |
| | MPA | 10 | 90.8 | 83.8 | 87.2 | 93.1 | 86.9 | 89.9 | 86.5 | 94.5 | 90.3 | 89.1 |
| | MPA | 20 | 90.4 | 87.2 | 88.8 | 93.0 | 89.9 | 91.4 | 89.6 | 94.2 | 91.9 | 90.7 |
| | MPA | 40 | 90.9 | 88.8 | 89.8 | 93.5 | 91.5 | 92.5 | 91.2 | 94.3 | 92.7 | 91.7 |
| | MPA | all | 91.6 | 82.3 | 86.7 | 94.8 | 88.3 | 91.4 | 92.3 | 93.6 | 93.0 | 90.4 |
| | CMPA | 5 | 93.3 | 80.1 | 86.2 | 94.2 | 87.0 | 90.5 | 85.1 | 94.9 | 89.8 | 88.8 |
| | CMPA | 10 | 90.8 | 88.1 | 89.4 | 93.4 | 89.8 | 91.6 | 90.8 | 93.7 | 92.2 | 91.1 |
| | CMPA | 20 | 90.8 | 86.5 | 88.6 | 93.7 | 90.0 | 91.8 | 91.7 | 93.7 | 92.7 | 91.0 |
| | CMPA | 40 | 90.9 | 84.8 | 87.7 | 94.1 | 89.2 | 91.6 | 92.2 | 93.5 | 92.8 | 90.7 |
| | CMPA | all | 91.7 | 82.2 | 86.7 | 94.8 | 88.3 | 91.4 | 92.3 | 93.6 | 92.9 | 90.4 |
| | MV | - | 96.0 | 63.9 | 76.7 | 95.7 | 78.7 | 86.4 | 77.1 | 94.9 | 85.1 | 82.7 |
| | Stanford | - | 65.4 | 62.4 | 63.8 | 78.9 | 76.1 | 77.5 | 78.4 | 85.2 | 81.7 | 74.3 |
| Singletons excluded | MPA | 5 | 93.6 | 76.0 | 83.9 | 90.1 | 57.9 | 70.5 | 71.3 | 59.7 | 65.0 | 73.1 |
| | MPA | 10 | 91.3 | 83.8 | 87.4 | 87.0 | 66.9 | 75.7 | 73.4 | 70.8 | 72.1 | 78.4 |
| | MPA | 20 | 90.8 | 88.4 | 89.6 | 86.2 | 75.8 | 80.7 | 76.7 | 77.7 | 77.2 | 82.5 |
| | MPA | 40 | 91.3 | 90.1 | 90.7 | 87.0 | 80.1 | 83.4 | 79.4 | 80.9 | 80.1 | 84.7 |
| | MPA | all | 92.2 | 89.2 | 90.7 | 88.1 | 77.8 | 82.6 | 79.5 | 80.2 | 79.8 | 84.4 |
| | CMPA | 5 | 93.6 | 81.1 | 86.9 | 89.4 | 66.5 | 76.3 | 75.5 | 65.3 | 70.0 | 77.7 |
| | CMPA | 10 | 91.1 | 88.8 | 89.9 | 85.9 | 76.2 | 80.8 | 78.2 | 77.7 | 78.0 | 82.9 |
| | CMPA | 20 | 91.3 | 89.5 | 90.4 | 86.7 | 78.8 | 82.6 | 79.1 | 80.2 | 79.7 | 84.2 |
| | CMPA | 40 | 91.6 | 90.1 | 90.9 | 87.3 | 78.5 | 82.6 | 80.0 | 80.1 | 80.1 | 84.5 |
| | CMPA | all | 92.4 | 89.1 | 90.7 | 88.1 | 77.8 | 82.6 | 79.4 | 80.0 | 79.7 | 84.3 |
| | MV | - | 96.1 | 64.8 | 77.4 | 93.8 | 45.0 | 60.8 | 66.3 | 48.5 | 56.1 | 64.8 |
| | Stanford | - | 65.7 | 62.1 | 63.9 | 50.3 | 42.5 | 46.1 | 42.7 | 49.8 | 46.0 | 52.0 |

Table 4: The quality of the silver chains evaluated using standard coreference metrics against gold chains. The "#" column shows the maximum user workload where the "all" configuration uses the dataset as it is (does not alter the player workloads). CMPA is shorthand for COMMUNITYMPA, MV for majority voting, while Stanford is the deterministic coreference system developed by Lee et al. (2011).

| | Method | MUC | | | | BCUB | | | CEAFE | | | Avg. F1 |
|---|---|---|---|---|---|---|---|---|---|---|---|---|
| | | # | P | R | F1 | P | R | F1 | P | R | F1 | |
| Singletons included | MPA | 5 | 81.3 | 62.8 | 70.9 | 73.7 | 65.2 | 69.2 | 64.4 | 76.7 | 70.0 | 70.0 |
| | MPA | 10 | 80.5 | 66.9 | 73.1 | 73.6 | 67.7 | 70.5 | 66.4 | 75.8 | 70.8 | 71.5 |
| | MPA | 20 | 78.1 | 73.2 | 75.6 | 68.4 | 70.9 | 69.6 | 68.3 | 73.6 | 70.9 | 72.0 |
| | MPA | 40 | 78.3 | 71.7 | 74.8 | 71.3 | 69.0 | 70.1 | 69.3 | 74.6 | 71.9 | 72.3 |
| | MPA | all | 79.3 | 72.5 | 75.7 | 72.1 | 69.3 | 70.7 | 70.5 | 73.2 | 71.8 | 72.7 |
| | CMPA | 5 | 80.2 | 67.1 | 73.0 | 72.8 | 66.7 | 69.6 | 67.5 | 75.6 | 71.3 | 71.3 |
| | CMPA | 10 | 78.8 | 72.7 | 75.6 | 72.4 | 69.5 | 70.9 | 70.1 | 73.9 | 71.9 | 72.8 |
| | CMPA | 20 | 76.6 | 73.9 | 75.2 | 70.9 | 69.3 | 70.1 | 70.7 | 73.3 | 72.0 | 72.4 |
| | CMPA | 40 | 78.8 | 72.2 | 75.3 | 71.2 | 70.0 | 70.6 | 69.8 | 74.1 | 71.9 | 72.6 |
| | CMPA | all | 78.9 | 71.0 | 74.7 | 71.8 | 68.0 | 69.9 | 69.6 | 73.6 | 71.6 | 72.0 |
| Singletons excluded | MPA | 5 | 81.3 | 62.8 | 70.9 | 60.4 | 37.3 | 46.1 | 52.7 | 37.4 | 43.7 | 53.6 |
| | MPA | 10 | 80.5 | 66.9 | 73.1 | 62.2 | 44.9 | 52.2 | 54.3 | 43.4 | 48.2 | 57.8 |
| | MPA | 20 | 78.1 | 73.2 | 75.6 | 51.5 | 54.4 | 52.9 | 55.5 | 47.8 | 51.4 | 60.0 |
| | MPA | 40 | 78.3 | 71.7 | 74.8 | 56.8 | 49.9 | 53.1 | 58.1 | 50.4 | 54.0 | 60.7 |
| | MPA | all | 79.3 | 72.5 | 75.7 | 58.3 | 52.4 | 55.2 | 58.3 | 49.5 | 53.5 | 61.5 |
| | CMPA | 5 | 80.2 | 67.1 | 73.0 | 58.4 | 42.9 | 49.5 | 55.2 | 43.2 | 48.5 | 57.0 |
| | CMPA | 10 | 78.8 | 72.7 | 75.6 | 58.5 | 51.5 | 54.8 | 56.2 | 49.4 | 52.6 | 61.0 |
| | CMPA | 20 | 76.6 | 73.9 | 75.2 | 56.4 | 51.3 | 53.7 | 54.6 | 50.2 | 52.4 | 60.4 |
| | CMPA | 40 | 78.8 | 72.2 | 75.3 | 56.2 | 52.5 | 54.3 | 55.9 | 49.5 | 52.5 | 60.7 |
| | CMPA | all | 78.9 | 71.0 | 74.7 | 57.2 | 49.1 | 52.8 | 56.3 | 47.7 | 51.7 | 59.7 |

Table 5: Detailed evaluation results of a state of the art coreference system trained on silver chains. The "#" column shows the maximum user workload where the "all" configuration uses the dataset as it is (does not alter the player workloads). CMPA is shorthand for COMMUNITYMPA.

| Method | Max. workload | P | R | F1 |
|---|---|---|---|---|
| MPA | 5 | 73.3 | 5.1 | 9.6 |
| MPA | 10 | 76.6 | 16.7 | 27.5 |
| MPA | 20 | 71.4 | 30.2 | 42.5 |
| MPA | 40 | 67.2 | 37.2 | 47.9 |
| MPA | all | 55.2 | 54.0 | 54.6 |
| COMMUNITYMPA | 5 | 69.3 | 24.2 | 35.9 |
| COMMUNITYMPA | 10 | 66.9 | 45.1 | 53.9 |
| COMMUNITYMPA | 20 | 60.5 | 48.4 | 53.8 |
| COMMUNITYMPA | 40 | 56.4 | 51.2 | 53.7 |
| COMMUNITYMPA | all | 49.6 | 54.9 | 52.1 |

Table 6: Detailed non referring scores (expletives + predicative NPs) obtained by a state of the art coreference system trained on silver chains.

