# OpenReview forum: "A Mention-Pair Model of Annotation with Nonparametric User Communities"
_ICLR.cc/2020/Conference — Reject_

### Official Review · AnonReviewer2 · 2019-10-23
**Official Blind Review #2**

**Rating:** 3

**Review:**

This paper extends the unpooled mention pair model of annotation (MPA) (Paun et al., 2018b) with the hierarchical priors (e.g., mean and variance) on the ability of the annotators. The proposed method was evaluated on Phrase Detectives 2 corpus, which was annotated by players in a game with a purpose setting for coreference resolution. To control the sparsity of the dataset, the authors split annotations from larger player workloads into smaller batches, and assumed that each batch was produced by a different player. The experimental results show that, when the data is sparse, the proposed method (CommunityMPA) worked better than MPA on Phrase Detectives 2 corpus in terms of mention-pair accuracy, silver-chain quality, and the performance of the state-of-the-art method trained on the aggregated mention pairs. This paper also includes a discussion about the inferred community profiles. The comparison with the traditional approaches that consider communities showed that the proposed method is comparable to the traditional approaches.

I am wondering of the connection between community and sparsity. This study assumes that knowledge of communities (spammers, adversarial, biased, average or high-quality players) would allow to regularize the ability of the annotators towards. In P2, this paper wrote, "This partially pooled structure can prove effective in conditions of sparsity where there are not enough observations to accurately estimate the ability of the annotators in isolation." I have two questions here: why is it effective to consider community for reducing the problem of the sparsity? If the knowledge of communities is useful, why did the advantage of the proposed method disappear in Figures 1, 2, and 3 when we used all data?

In addition, I'm not convinced with the idea of "breaking the larger player workloads into smaller batches" for simulating the sparsity and communities. This treatment introduces quite a few shadow users whose capabilities are exactly the same, and deviates from the reality of the user community. Does this treatment favor the proposed method over MPA more than necessary? I am also wondering why evaluating portions of the dataset where annotations were made by 'sparse' users would not work to highlight the effectiveness of the proposed method for sparse users.

The impact of this paper would be greater if the experimental results could support the importance of modeling user community on the real data. The authors may justify the simulation procedure of the sparsity because 98.67% of Phrase Detectives 2 corpus was annotated by those who produced more than 40 annotations. However, I also think that it is important to show how the proposed method is effective on the real data with sparse annotators. Currently, Table 1 showed no improvement over the conventional methods.

Minor comment

In Section 2.1: It was difficult to separate which part is the base model (MPA) and the novel proposal without reading Paun et al. (2018b).

**Experience Assessment:**

I have read many papers in this area.

**Review Assessment: Checking Correctness Of Derivations And Theory:**

I assessed the sensibility of the derivations and theory.

**Review Assessment: Checking Correctness Of Experiments:**

I carefully checked the experiments.

**Review Assessment: Thoroughness In Paper Reading:**

I read the paper thoroughly.

---

> ### Author Response · Authors · 2019-11-07
> **Detailed response to Official Blind Review #2**
>
> We wish to thank the reviewer for the detailed reading of the paper.
>
> Concern
>
> Why is it effective to consider community for reducing the problem of the sparsity?
>
> Answer
>
> A community is simply a hierarchical structure. Such structures are effective in alleviating sparsity because they pool the estimates of the lower level parameters towards their mean; this is known in statistics as shrinkage. Sparsity is alleviated because you now have more observations informing the lower parameters compared to an unpooled model. We’ll explain this effect in the context of our model. In an unpooled structure such as MPA’s  the annotations of an annotator inform only his ability. When the data is sparse, it may be difficult for MPA to properly profile the ability of the annotators simply because there not enough observations available to get a good estimate. In CommunityMPA, however, the annotations of an annotator influence both the estimates of his ability and, indirectly, the estimates of the other annotators that are part of the same community (the annotator’s annotations influence his ability which informs the overall ability of the community which influences in turn the ability of all its members).
>
> Concern
>
> If the knowledge of communities is useful, why did the advantage of the proposed method disappear in Figures 1, 2, and 3 when we used all data?
>
> Unpooled models work well when there is sufficient data. The advantage of  hierarchical models over unpooled ones is their versatility - their ability to work better in conditions of data sparsity, while achieving a similar level of performance  when there is sufficient data. This is because the community structure acts as a prior on the individual annotator estimates; when plenty of annotations are available, their likelihood simply dominates the effects of the community prior. In unpooled models, the posterior of the annotator estimate is influenced only by the annotations' likelihood (and a fixed prior); thus, as the number of annotations grows, the performance of the unpooled models converges on that of hierarchical ones. The figures in question show this expected behavior.
>
> We will make this point clearer in the paper and therefore that the observed behavior is exactly what is to be expected.
>
> Concern
>
> Does this treatment favor the proposed method over MPA more than necessary? I am also wondering why evaluating portions of the dataset where annotations were made by 'sparse' users would not work to highlight the effectiveness of the proposed method for sparse users.
>
> Answer
>
> What we are claiming is that CommunityMPA simply covers a wider range of crowdsourcing scenarios than MPA. This aspect is backed theoretically by the wider adaptability of the hierarchical structures over the unpooled ones in conditions of varying sparsity. We demonstrate this practically in the crowdsourcing setups we introduced in the paper.
>
> There are ‘sparse’ users in the PD dataset but those are accompanied by users with large workloads (a few users managed to annotate most of the dataset). This essentially gives the unpooled model (MPA) a reliable source of adjudication even for the items annotated by ‘sparse’ users.
>
> Concern
>
> "The impact of this paper would be greater if the experimental results could support the importance of modeling user community on the real data. The authors may justify the simulation procedure of the sparsity because 98.67% of Phrase Detectives 2 corpus was annotated by those who produced more than 40 annotations. However, I also think that it is important to show how the proposed method is effective on the real data with sparse annotators. Currently, Table 1 showed no improvement over the conventional methods."
>
> Answer
>
> We agree that an evaluation on real sparse datasets would increase the impact of the paper; the problem is that although coreference annotation  (and other related tasks such as entity linking) are becoming increasingly important for the NLP community, so far there have been only limited attempts to do this annotation via crowdsourcing,  and therefore to our knowledge  there are no other publicly  available resources that require the annotation scheme assumed by our model. Even for traditional classification tasks, the state of the art work, Moreno et al. 2015, used in part completely synthetic data to prove the benefits of the model sparsity-wise. We tried to avoid using completely synthetic data because of concerns similar to those expressed by the reviewer, but we had to induce the different levels of sparsity required to test our hypothesis in some way.

---

### Official Review · AnonReviewer1 · 2019-10-26
**Official Blind Review #1**

**Rating:** 8

**Review:**

The paper proposes an improvement over the mention-pair model for anaphoric annotation by Paun et al. to mitigate the effects of sparsity that are inherently present in some crowd-sourcing environments. The extension involves hierarchic communities of annotators, where the number of communities grows with data via "Dirichlet process mixture based on a stick-breaking representation of the underlying Dirichlet process".

The paper is very well-written and the results are convincing. Experiments pack appropriate breadth of comparison and the discussion is relatively extensive. THe coverage of related work is extensive and the contribution placement apt. The paper does not state the language(s); it seems to be English-only, and it is not clear how the model would scale up/down with the number of languages or limited availability of resources.

I see no major issues with accepting the paper.

**Experience Assessment:**

I do not know much about this area.

**Review Assessment: Checking Correctness Of Derivations And Theory:**

I assessed the sensibility of the derivations and theory.

**Review Assessment: Checking Correctness Of Experiments:**

I assessed the sensibility of the experiments.

**Review Assessment: Thoroughness In Paper Reading:**

I made a quick assessment of this paper.

---

> ### Author Response · Authors · 2019-11-07
> **Detailed response to Official Blind Review #1**
>
> Thank you for the review.
>
> Concern
>
> The paper does not state the language(s); it seems to be English-only, and it is not clear how the model would scale up/down with the number of languages or limited availability of resources.
>
> Answer
>
> The annotated texts are in English, but the model is language independent and appropriate for any task where the annotation labels would belong to some hierarchical classes. In our case we dealt with anaphoric labels (e.g., the most recent antecedents of a mention) which belong to information status classes (e.g., discourse old, discourse new), but the approach would also apply, e.g., to entity linking.

---

### Official Review · AnonReviewer4 · 2019-10-30
**Official Blind Review #4**

**Rating:** 6

**Review:**

This paper presents a method to get a more qualified output from multiple crowdsourced mention pair annotations for coreference annotation. It builds on Paun et al. (2018b) who presented a probabilistic model able to aggregate crowdsourced data while also turning co-reference annotation into a task with four pre-determined classed, denoted the Mention Pair Model (MPA). This study extends MPA to CommunityMPA by including information about the annotators' hierarchical community profiles from Simpson et al. (2011, 2013): spammers, adversarial, biased, average, and high-quality players. CommunityMPA mention pairs are compared to MPA and silver chains derived from both mention pairs are compared to gold chains. The paper also compares a SOTA model trained on silver chains. The general trend for all experiments is that for lower annotator workloads, CommunityMPA is better, but performance for both models generally increase with higher annotator workloads. For all experiments, using higher annotator workload (40 or higher) is the better and MPA is then better or on par. MPA and CommunityMPA are also compared to results from Moreno et al. (2015) on other tasks and datasets showing on par or better performance.

The paper is well-written and the method is well-justified and generally well-described. The comparisons are meaningful. The work in interesting and much needed if SOTA indeed is from 2015, but the CommunityMPA results are not strong enough for me to recommend full accept.

Strengths:
This seems like a simple, strong and sensible approach when annotations per annotator are sparse.

Weaknesses:
But performance is not better than MPA with a higher workload. When designing the annotation process, it does not seem like a good idea to use CommunityMPA over asking for a minimum of 40 annotations per annotator and use MPA.

Questions:
Which state of the art coreference system do you use for results in Figure 3?

Small comments
Figure 4 is an interesting insight into the four community profiles (but why not include adversaries?) but is kind of detached from the rest of the experiments, since these profiles are not discussed much in the remaining part of the paper. I suggest explaining community profiles more.

Table 1: I would find it useful if you would briefly introduce the tasks from Moreno et al. (2015). Otherwise, I would assume the task to be the same as previous parts of the paper. The number of correctly labelled items seems irrelevant, whereas the size of each dataset is relevant to include (though not necessarily in the table)
p 7: seminar work -> seminal work?
Caption of Table 1: (Moreno et al., 2015) > Moreno et al. (2015)
Table 1: Boldface best result per dataset

**Experience Assessment:**

I have read many papers in this area.

**Review Assessment: Checking Correctness Of Derivations And Theory:**

I assessed the sensibility of the derivations and theory.

**Review Assessment: Checking Correctness Of Experiments:**

I assessed the sensibility of the experiments.

**Review Assessment: Thoroughness In Paper Reading:**

I read the paper thoroughly.

---

> ### Author Response · Authors · 2019-11-07
> **Detailed response to Official Blind Review #4**
>
> We wish to thank the reviewer for the detailed reading of the paper. We address the main concerns in order.
>
> Concern
>
> But performance is not better than MPA with a higher workload. When designing the annotation process, it does not seem like a good idea to use CommunityMPA over asking for a minimum of 40 annotations per annotator and use MPA.
>
> Answer
>
> Each annotation project is subject to some constraints. Indeed, the solution proposed by the reviewer could  be  adopted if feasible within the constraints of the project (e.g., if the annotators can each do enough annotations). But in case this is not possible, using CommunityMPA would maximise the achievable quality. CommunityMPA is a hierarchical model where the level of pooling exerted by the communities over the individual annotator abilities is as strong as evidenced by the data. Hierarchical models have the advantage over their unpooled counterparts that they can share information to the lower level parameters through the hierarchical structure, thus being able to alleviate the effects of sparsity. When enough observations are available a hierarchical model should provide similar estimates to those of an unpooled model. Going back to our paper, the communities act as priors on the ability of the annotators; the posterior of the latter thus depends on the community information and the observed annotations which, when a large number of them is available, will dominate this posterior. For an unpooled model like MPA the posterior of the annotator estimates depends on some fixed prior and the annotations. Having said that, the two cases of posterior estimates should be similar when enough observations are available.
>
> Concern
>
> Which state of the art coreference system do you use for results in Figure 3?
>
> Answer
>
> We use the cluster ranking model from Poesio et al, 2019 (https://www.aclweb.org/anthology/N19-1176.pdf see Section 6), which provides state of the art performance on the Phrase Detectives dataset. It was specifically developed to accommodate the richer annotation scheme of this corpus, which, unlike Ontonotes, also includes singletons and non-referring expressions. It should  be noted that  the model overperforms the known Lee et al. (2018) model (state of the art on Ontonotes up until recently) when singletons are excluded (setting in which the two models can be compared).
>
> This information is briefly mentioned in the text (Section 3.2). We’ll update also the caption of Figure 3 to point to Poesio et al, 2019.
>
> Concern
>
> Table 1: I would find it useful if you would briefly introduce the tasks from Moreno et al. (2015). Otherwise, I would assume the task to be the same as previous parts of the paper. The number of correctly labelled items seems irrelevant, whereas the size of each dataset is relevant to include (though not necessarily in the table)
>
> Answer
>
> No, the tasks used by Moreno et al are different. We will make this point clearer and add a description of these tasks in the final version of the paper if accepted.

---

> > ### Comment · AnonReviewer4 · 2019-11-14
> > **Thanks. I read your rebuttal**
> >
> > I am happy that you will add these clarifications in the paper. It does not change the overall score.

---

### Official Review · AnonReviewer3 · 2019-10-31
**Official Blind Review #3**

**Rating:** 6

**Review:**

The paper presents an interesting crowdsourcing approach as an alternative to expert annotation. Different from the conventional crowdsourcing approaches that concentrate on classification tasks with predefined classes, the proposed Community MPA, as an extension of MPA can address anaphoric annotation well where coders relate markables to coreference chains whose number cannot be predefined. Authors demonstrate that on a recent large-scale crowdsourced anaphora dataset, the proposed Community MPA works better than the unpooled counterpart in conditions of sparsity, and on par when enough observations are available. It also generalizes well to other crowdsourcing tasks under different settings.

* The paper is an incremental work to the existing one by Paun et al. (2018b). It considers the underlying communities of different workers, which may lead to better annotations when the data is scarce.
* It is also partially similar to the Moreno et al. (2015) that models a latent number of communities through Gibbs sampling. The proposed method also shows comparable performance compared to Moreno et al. (2015) in standard annotation tasks.
* Technically, the paper is written well (I appreciate detailed deduction in the appendix), but is hard to follow for people not from the crowdsourcing field. There are lots of field-specific terms without much explanation. It may take quite a while for readers with only machine learning background.
* The complexity of the work is not explicitly explained. While the solution itself refers to (Blei et al., 2017), it would be better to show running time in experiments.
* Authors claimed that “We let the number of communities grow with the data, a flexibility that we achieve using a Dirichlet process mixture,” but this is not clear in the modeling in section 2.1. Which parameters control such increases and how they were adjusted? More explanations are needed.

**Experience Assessment:**

I do not know much about this area.

**Review Assessment: Checking Correctness Of Derivations And Theory:**

I assessed the sensibility of the derivations and theory.

**Review Assessment: Checking Correctness Of Experiments:**

I carefully checked the experiments.

**Review Assessment: Thoroughness In Paper Reading:**

I read the paper at least twice and used my best judgement in assessing the paper.

---

> ### Author Response · Authors · 2019-11-07
> **Detailed response to Official Blind Review #3**
>
> We wish to thank the reviewer for the comments.
>
> Concern
>
> The paper is an incremental work to the existing one by Paun et al. (2018b).
>
> Answer
>
> The paper does build on Paun et al 2018, but the extension involves the development of a completely new model:
>
> - Model specification:  CommunityMPA  is a nonparametric partially pooled model, fitting jointly with the ability of the annotators hierarchical community profiles. In order to make the number of communities adapt  to and grow  with the data we had to use  a completely new formalization involving  a Dirichlet process mixture based on a stick-breaking representation of the underlying Dirichlet process.
>
> - Parameter estimation:  Both CommunityMPA and the original MPA have their parameters estimated using variational inference, but  the inference process required for the new model  is substantially  more involved. MPA is a straightforward application of variational inference for models whose complete conditionals are in the exponential family. CommunityMPA  makes use as well of this elegant theory, but it also has to: 1) deal with cases of non-conjugacy (caused by the standard logistic function; see the generative model in the paper), and 2) use truncated variational distributions to approximate the unbounded mixture of communities.
>
> Concern
>
> Authors claimed that “We let the number of communities grow with the data, a flexibility that we achieve using a Dirichlet process mixture,” but this is not clear in the modeling in section 2.1. Which parameters control such increases and how they were adjusted? More explanations are needed.
>
> Answer
>
> We thank the reviewer for pointing out this omission. In the final version of the paper, if accepted, we will explain how  the stick-breaking construction works and how it achieves that.

---

### Decision · Program_Chairs · 2019-12-19

**Decision:**

Reject

**Comment:**

Thanks to the reviewers and the authors for an interesting discussion. The reviewers are mixed, learning toward positive, but a few shortcomings were left unaddressed: (i) Turning the task into a mention-pair classification problem ignores the mention detection step, and synergies from joint modeling are lost. (ii) Lee et al. (2018) has been surpassed by some margin by BERT and spanBERT, models ignored in this paper. (iii) Several approaches to aggregating structured annotations have already been introduced, e.g., for sequence labelling tasks. [0] Overall, the limited novelty, the missing baselines, and the missing related work lead me to not favor acceptance at this point.

[0] https://www.aclweb.org/anthology/P17-1028/